# What Matters in Branch Specialization?
# Using a Toy Task to Make Predictions

## Abstract

What motivates the brain to allocate tasks to different regions and what distinguishes multiple-demand brain regions and the tasks they perform from ones in highly specialized areas? Here we explore these neuroscientific questions using a purely computational framework and theoretical insights. In particular, we focus on how branches of a neural network learn representations contingent on their architecture and optimization task. We train branched neural networks on families of Gabor filters as the input training distribution and optimize them to perform combinations of angle, average color, and size approximation tasks. We find that networks predictably allocate tasks to the branches with appropriate inductive biases. However, this task-to-branch matching is not required for branch specialization, as even identical branches in a network tend to specialize. Finally, we show that branch specialization can be controlled by a curriculum in which tasks are alternated instead of jointly trained. Longer training between alternation corresponds to more even task distribution among branches, providing a possible model for multiple-demand regions in the brain.

## 1 Introduction

Brain specialization has been an active topic in neuroscience for decades, and has helped us discover many brain regions dedicated to particular tasks across humans despite low-level differences in plasticity (*eg* the Fusiform Face Area (Kanwisher et al., 1997) that serves a pivotal role for face recognition; though also see Gauthier et al. (1999); Arcaro et al. (2017, 2020); Hesse & Tsao (2020)). However, we have yet to build a full explanation of why some "multiple-demand" brain systems (Fedorenko et al., 2013) are involved in a variety of tasks while others are extremely narrow in scope. We do not currently know all the factors that distinguish general and specialized brain regions, nor what causes them to emerge or develop; in addition, they are challenging to study *in vivo* because biological network architectures cannot be easily modified and paired with the proper controls. Consequently, we turn to deep learning, where computational cognitive neuroscientists may now test their ideas on computer models rather than living organisms and can also play the role of a "virtual neurophysiologist" by inspecting artificial neural network activations (Zeiler & Fergus, 2014; Kriegeskorte, 2015; Olah et al., 2020; Hamblin & Alvarez, 2021).

Indeed, the field of machine learning has shown interest in neural network specialization: Voss et al. (2021) at OpenAI showed that branching still occurs implicitly in neural networks even when the network architecture does not explicitly bifurcate. On the neuroscience side, examples of previous works on branching in audition include Kell et al. (2018) where a branched neural network jointly trained to learn speech and music learned to correlate well with human auditive behaviour. And in vision, Dobs et al. (2021) found that jointly training networks on a face and object classification task resulted in specialized branching along the hierarchy of the CNNs.

Submitted to 3rd Workshop on Shared Visual Representations in Human and Machine Intelligence (SVRHM 2021) of the Neural Information Processing Systems (NeurIPS) conference.

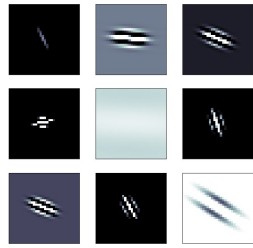

Figure 1: Sample images generated for the Gabor dataset. See Appendix A for details.

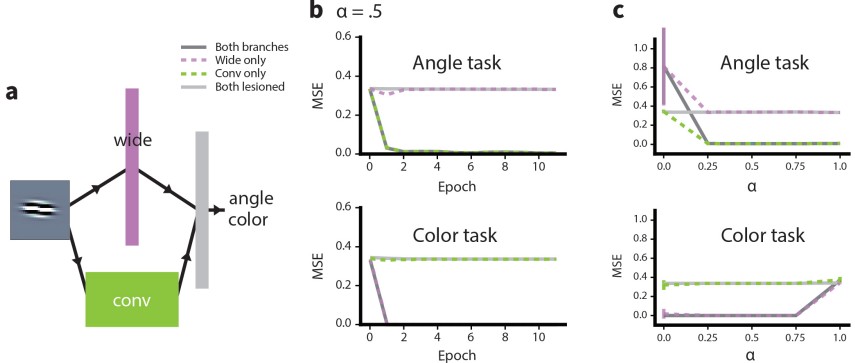

Figure 2: **a.** A branched network with one fully connected wide branch and a convolutional branch. Outputs are the angle and color of the $32 \times 32$ Gabor filter image. **b.** Mean squared errors (MSEs) of branched network during training in four conditions: intact, with the wide branch only, with the convolutional branch only, and with both branches lesioned. Branches specialize early on in training and do not appear to change afterward. **c.** Statistics of MSEs for ten random seeds for the experiment in **b**, for different values of $\alpha$ (see text). Standard deviations shown in error bars. Task weights as defined by $\alpha$ do not affect branch specialization.

Here we hope to complement observations from the previous studies, and begin identifying the factors involved in branch specialization. In particular we focus on specialization in a precise visual task where the stimuli are Gabor patches (Fig. 1) rather than natural images such as those of ImageNet (Russakovsky et al., 2015), VGGFace2 (Cao et al., 2018), THINGS (Hebart et al., 2019) or Places (Zhou et al., 2017). The benefit of Gabors is that we can design tasks that need not be compositional or hierarchically-local such as object recognition, but can be global (*e.g.* average color/luminance) and/or order-1 hierarchically local (*e.g.* orientation) (Deza et al., 2020). And in contrast to most prior work, we change our tasks, architectures, and training protocols to try to unravel the *causal factors* for branches to specialize rather than only observing this phenomena after training.

The rest of the paper is organized as follows. In section 2, we show that branch specialization is robust and happens with diverse or identical branches. In the case of branches with different architectures trained on two simultaneous tasks where each is better suited for only one of the branches, a network specializes in predictable ways that align with the branches' inductive biases. In section 3, we show that branch specialization can be controlled by a curriculum learning scheme that alternates task training, and that the faster the alternation rate, the more likely specialization is to occur. In the discussion, we note some neuroscientific implications of our results and describe planned future work on both the mechanisms of branch specialization and its consequences.

## 2 Task allocation in branches can be predicted by inductive biases

This section asks how consistently branch specialization occurs, and how much architectural biases affect it. In Figure 2, we used a Gabor filter dataset and asked our networks to simultaneously output the images' angle and average color. Our network architecture consisted of two branches (see Figure 2a). One was a convolutional network with two convolutional layers followed by two fully connected layers (see Appendix for details). The other branch was a fully connected network with two fully connected layers. The outputs of both branches are then fed into a linear output layer, from which the two output values for the dual task (angle and color) are read. See the associated code for implementation.[1] We expected the angle task to be better suited to the inductive biases of the convolutional branch, perhaps requiring edge detection or similar computations that convolutions can more easily learn. Conversely, mean color estimation is a simple average that is better suited to the fully connected branch. In this way, we have designed each branch of the network to have an inductive bias that matches only one of the tasks. In our experiments, however, the fully connected branch has almost three times fewer parameters than the convolutional branch (see Appendix B.2.1 for details), so in one sense, it may be surprising if the fully connected branch learns a task at all.

Results are shown in Figure 2b and c. We train the entire network on the dual task but then evaluate branches individually by zeroing out (*i.e.* lesioning) the final outputs of the other branch before it is fed into the linear output layer. We compare those results to ones where both branches were lesioned and where the network is intact. Figure 2b shows that training converges quickly and each task is entirely localized to one branch. Over ten random seeds for the training in Figure 2b, an intact network had an average MSE of .0074(.0044) on the angle task (standard deviation in parentheses). The fully connected branch alone had an average MSE of .3367(.0045) on the angle task, whereas the convolutional branch alone had an average MSE of .0074(.0044). With both branches lesioned, the average MSE was .3367(.0045). We can conclude, then, that the convolutional branch is responsible for the angle task. These data are shown in Figure 2c. Figure 2c also tells a similar story for the color estimation task, except that the fully connected network is now responsible for it instead. In ten random seeds, the angle task was always localized to the convolutional branch while color estimation was always localized to the fully connected branch. These allocations align precisely with the inductive biases we initially described.

We then wanted to see if branch specialization was robust to the two tasks' relative contributions to loss, so we scaled the losses for both tasks with a convex-combination parameter $\alpha$ to define a new loss $\mathcal{L} = \alpha\mathcal{L}_{angle} + (1-\alpha)\mathcal{L}_{color}$. We tried a range of $\alpha$ values from 0 to 1 with the same network architectures in Figure 2a and the same dual task. One might expect that if one task's relative importance were to increase, a network may allocate more resources to it rather than continuing to split resources evenly. However, we did not see any gradual change in resource allocation. Rather, we saw the same branch specializations as before regardless of task importance, even for losses that were heavily biased toward one task. Furthermore, even when the branched network was trained on only the color or only the angle task (corresponding to $\alpha = 0$ and $\alpha = 1$ respectively), one branch was left unused while other shouldered the entire task burden. Thus, relative contributions of each task to the loss did not affect task allocation to branches.

Next, we wanted to know what would happen with two identical branches and two more similar tasks. If branches didn't have asymmetrical biases for tasks, would they still exhibit branch specialization? We used the architecture described in Figure 3a with two convolutional network branches this time, with all else the same as the network in Figure 2a. For training we used another simultaneous dual task setup where the input was a Gabor filter and the output was two values: angle and size, where size was set by the parameter $\omega$ in the Gabor filter generation function (see Appendix).

We plot training progress in Figure 3b. As before, branches quickly specialize to one task. Because the branches were identical, we used their effect on the size task to label them "conv 1" or "conv 2" in all of Figure 3 and ordered the branches by their performance on the size task to maintain functional identity. Tasks are consistently allocated to different branches over five random initializations, although in the angle task, both branches were occasionally involved. However, one branch was always more important: in an intact network with $\alpha = .5$, the average MSE was .0079(.0036) for the angle task and with "conv 1" it was .0498(.0832). "Conv 2" had a significantly larger MSE of .2724(.1229) while lesioning both branches only increased that number to .3381(.0022). We

---

[1]Code is available at https://github.com/ccli3896/branches-svrhm.

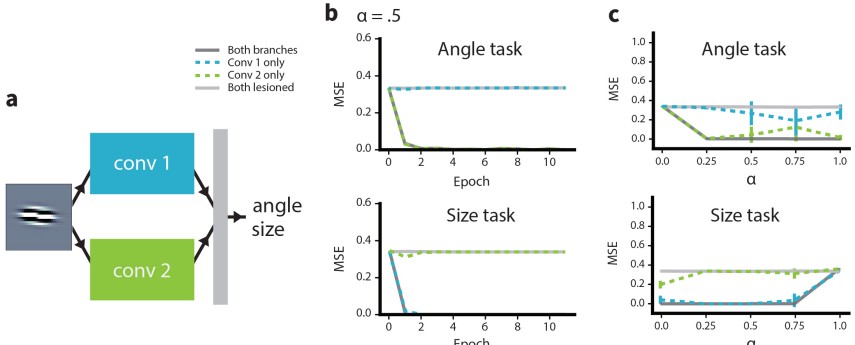

Figure 3: **a.** Branched network architecture with two identical convolutional branches. In this figure branches are distinguished by functionality after training. "Conv 1" is more important for performance in the size task than "conv 2." Outputs are now angle and size of Gabor filters. **b.** Training progress on the simultaneous angle/size task. Plots show performance during training of MSEs for the intact network, "conv 1 only", "conv 2" only, or both branches lesioned. Again, specialization happens early. **c.** MSE statistics over five random seeds for different values of $\alpha$ (see text), with standard deviations in error bars. Complete task separation does not always happen as in Figure 2, but there is clear specialization particularly for the size task.

tried different relative task weightings as in Figure 2c, again showing that our results were not too sensitive to the loss function. Overall, branch specialization happened even with more similar tasks and identical branch architectures, so although specialization can be predicted by matching inductive biases to tasks, it is not dependent on task-branch asymmetry.

## 3   Branch specialization can be controlled by curriculum learning

In this section we ask whether branch specialization can be controlled. Looking at individual networks' branch allocations from the experiments for Figure 3c (not shown), we noticed that branched networks trained on one task were more likely to use both branches for it. We hypothesized that alternating between tasks could lead to task sharing between branches rather than specialization.[2] To test the hypothesis, we trained the branched architecture in Figure 4a on the same Gabor angle and size task as in Figure 3. This time, losses came entirely from one task for $n$ epochs and switched to the other task for the next $n$ epochs, alternating for 500 epochs. We tested 1, 5, 10, and 20 for values of $n$.

For small values of $n$, training is more like the simultaneous dual task and we expect branch specialization to happen. Our results for $n = 1$, or task alternation between every epoch, are shown in Figure 4a. Aside from some epochs where EWC (see footnote) fails to maintain task performance as seen in the spikes in Figure 4a-bottom, we observe consistent branch specialization for the angle and size tasks. The MSEs over five random seeds for the angle task, for instance, were .0065(.0017) for the intact network and .0065(.0017) when we lesioned the branch with worse performance on the angle task. When the branch with better performance was lesioned, MSEs were .3392(.0038), compared to both branches lesioned at .3392(.0037). Data for the size task are in Figure 4c.

When $n$ is increased to 10, we see more distribution of both tasks over both branches. Figure 4b displays the errors from trained networks with all combinations of branch lesions. Now, MSEs for five random seeds for the angle task were .0835(.0189) for the intact network and .1247(.0332) with the worse branch lesioned. With the better branch lesioned MSEs were .2453(.0330), compared to both lesioned with an average MSE of .3397(.0034). On both tasks, both branches have non-negligible contribution to task performance. Results for all tested values of $n$, the epochs between task alternation, are in Figure 4c. Faster alternation (corresponding to smaller values of $n$) more

---

[2]To prevent catastrophic forgetting, we used Elastic Weight Consolidation (Kirkpatrick et al., 2017), or EWC during training. EWC is inspired by synaptic weight consolidation in the brain, where synapses that are involved in long-term memories are strengthened and stay strong for days, weeks, or years (Clopath, 2012). In a similar way, when a network is trained on a new task after having already learned another, EWC prevents weights that are important to previous tasks from changing as much as unimportant weights.

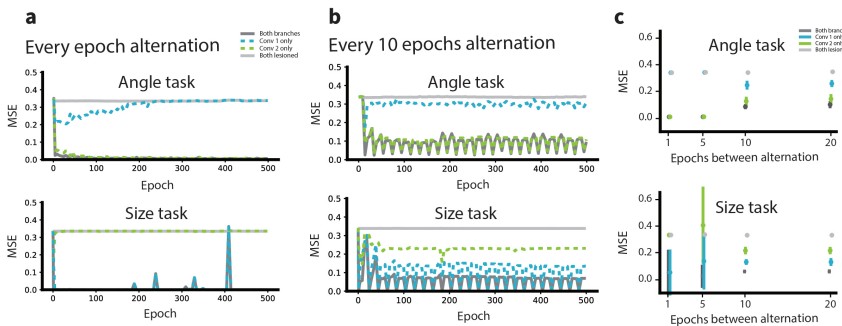

Figure 4: The effects of task alternation on branch specialization. **a.** and **b.** show training progress with intact networks, one branch only, or both branches lesioned. **a** is data from alternating between angle and size tasks every epoch and **b** alternates them every ten epochs. Complete branch specialization happens as usual in **a** but not in **b**, which distributes tasks more between the branches. **c.** shows MSEs for intact networks, one branch, and both branches lesioned over five seeds at the end of training, with standard deviations in error bars. More epochs between task switching decreases specialization, but further decreasing alternation frequency does not further reduce specialization in these experiments.

often led to branches that were entirely responsible for one of the two tasks. For slower alternation, both branches affect both tasks. However, one branch tended to be more important in each task, and this did not change as alternation times grew. Nonetheless, we could partially control the degree of branch specialization by changing the learning curriculum.

## 4   Discussion

Understanding how tasks distribute themselves within a neural network has relevance to machine learning research, perhaps to aid in architecture design, and to neuroscientists, to understand why parts of the brain can be highly specialized or be involved in a broad set of functions. We use small networks and a simple task set to try to gain intuitions for larger models and more complex tasks. As a consequence, our conclusions may serve best as guides for larger experiments in the future.

Using a toy Gabor filter dataset and dual task experiments, we show that branch specialization is a robust phenomenon with either identical or different branches. In the case of branches with different architectures that are better suited to one of a set of tasks, task allocation to branches can be predicted based on branches' inductive biases. We also demonstrate that specialization is sensitive to training curricula and if multiple tasks alternate during training, branch specialization can be reduced.

These results suggest some neuroscientific hypotheses about how and when branch specialization occurs. One can predict that brain regions that are consistently allocated to the same tasks across individuals have architectural inductive biases for those tasks, as mentioned by previous studies such as Dobs et al. (2021). Another prediction is that that the difference between multiple-demand and specialized brain regions may be dependent on the statistics of task presentation throughout an animal's life—that is, perhaps more functions in multiple-demand regions are blocked and infrequent compared to a task like visual perception, which the highly specialized human visual cortex must perform almost every waking hour.

For future work, we would like to better understand both the mechanisms leading to branch specialization and the consequences of it. In terms of mechanisms, training dynamics would be interesting to study, since based on how reliably we see branch specialization in small models and large vision models (Voss et al., 2021) alike, we expect the loss surfaces of branches within one architecture to interact with each other in a feedback loop that pushes branch functions away from each other during training. With regards to the consequences of branch specialization, we want to know when specialization is or is not useful. If both very specific and very broadly-used regions exist in the brain, does the distribution of neural substrate that these tasks are computed on affect performance? In the future we would like to understand this task localization trade-off, and its broader implications to generalization in human and machine visual intelligence.

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

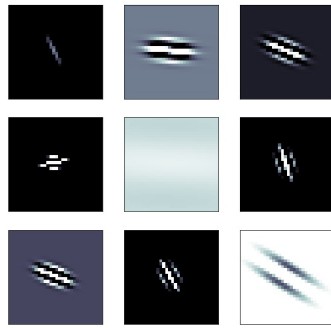

Figure 5: Sample images generated for the Gabor dataset.

Bolei Zhou, Agata Lapedriza, Aditya Khosla, Aude Oliva, and Antonio Torralba. Places: A 10 million image database for scene recognition. *IEEE transactions on pattern analysis and machine intelligence*, 40(6):1452–1464, 2017.

## A    Creation of Gabor Dataset

Gabor filters are defined by the following sets of equations. $\theta$ is the angle parameter mentioned in Figures 2, 3, and 4. $\omega$ is the size parameter in Figures 3 and 4.

$$g(x', y', \omega) = \frac{\omega^2}{4\pi^3} \exp\left(\frac{-\omega^2}{8\pi^2} * (4x'^2 + y'^2)\right) \exp(\pi^2/2) * \cos(\omega x')$$
$$x' = x\cos\theta + y\sin\theta$$
$$y' = -x\sin\theta + y\cos\theta$$

We then added a random number to the image from a uniform distribution $\in [-1, 1]$ to vary the image colors. For all experiments, $\theta \in [0, \pi]$ and $\omega \in [.1, 3]$, both drawn randomly from uniform distributions. Image sizes were $32 \times 32$. The training set was 20k images and the test set was 10k images.

## B    Training networks

All code is available online at https://github.com/ccli3896/branches-svrhm.

### B.1    Hardware

All experiments use neural networks written in Pytorch (Paszke et al., 2019). Experiments were run on shared GPUs in Google Colab (with GPU models allocated from NVIDIA K80, T4, P4, and P100) and the FASRC cluster, supported by the FAS Division of Science Research Computing Group at Harvard University (containing automatically allocated GPU models from among NVIDIA K20m, K40m, K80, M40, 1080, TITAN X, TITAN V, P100, V100, and RTX2080TI).

## B.2 Network and training parameters

### B.2.1 Network sizes

All branched networks had the following traits: they took $32 \times 32$ image inputs fed immediately into two branches. The two branches' outputs were concatenated and fed into a linear readout layer, which always output two values.

Convolutional branches had two convolutional layers, max pooling, and two fully connected layers. The first convolutional layer took in one channel and output 32 channels with a kernel size of 3 and padding of 1. The second layer was the same except that the input was 32 channels as well. The max pooling layer had a kernel size of $2 \times 2$. It was followed by two dense layers. The first took an input size of 2048 (the flattened output of the max pool layer). Its output size was 120. The second dense layer took an input size of 120 and output of 84. All layers used relu as an activation function. In total the convolutional branch had 265612 trainable parameters.

The fully connected branch used in Figure 2 was a two-layer fully connected network with an input size of $32 \times 32$ (the flattened image). The first layer had an output size of 86; the second layer took an input of size 86 and output 10. Both layers used relu for an activation function. In total there were 89020 trainable parameters.

Thus, the networks with fully connected and convolutional branches had 265612 conv branch + 89020 dense branch + (84 conv outputs + 10 dense outputs) ∗ 2 outputs + 2 biases for outputs = 354822 parameters. Networks with two convolutional branches had $265612 * 2 + (84 * 2) * 2 + 2 = 531562$ parameters.

### B.2.2 Training

For training all models, we used Adam optimizer with a learning rate of .001. For experiments in Figure 4, we used dropout with a rate of .5 and elastic weight consolidation with an importance (see Kirkpatrick et al. (2017)) $1e4$ for the size task and $1e7$ for the angle task, both chosen after a period of tuning.

