# OpenReview forum: "What Matters In Branch Specialization? Using a Toy Task to Make Predictions"
_NeurIPS.cc/2021/Workshop/SVRHM — SVRHM 2021 Poster_

### Official Review · Reviewer_9w3b · 2021-10-23
**What Matters In Branch Specialization?**

**Rating:** 6
**Confidence:** 3

**Review:**

This paper examines the question of what causes branch specialization in deep neural nets, a phenomenon which has been observed empirically, most notably, in Voss et al. (2021). They find that task-to-branch specialization occurs in conditions where each branch is specialized for a given subtask, and when each branch is identical. They also find that specialization does not occur if subtasks are slowly interlaced.

I found the main result of this paper intriguing. The subject of this paper is still poorly understood, hence structured experiments will be necessary to advance our theoretical understanding. The paper was a bit hard to read, and I found the experiments not very conclusive. Some cleanup of the paper would make it easier to read.

**Things that could be improved**

*Abstract*

I think the framing in terms of MD vs. single-use areas is a bit strange. I tend to associate branches more with sub-compartments within an area (e.g. stripes in V2) or streams (e.g. ventral vs. dorsal). The last sentence (lines 13 to 15) was awkwardly phrased, it should be cleaned up.

*Intro*

On lines 19-20, there's a whole sentence with embedded references inside the parentheses, I think this should be its own sentence without the parentheses.

*Section 2*

I would have liked firmer ground to justify that one task was convolutional vs. and the other was easier to solve with a fully connected network. At face value, an average is easy to compute with convolutions.

After line 94, I think a subheading is deserved, e.g. Specialization emerges in symmetric branches. The numbers in the text could be put into a table.

*Figures*

The metric (MSE) is inverse scored (lower is better) but then lesions are used to compute these scores, hence a low score means a branch is *not* involved in Figures 2, 3, 4 - the double negative requires mental gymnastics, would prefer a less convoluted metric. The font is impossible small in some of the figures. Panel b is not very helpful in figures 2, 3.

Figure 4 panel C contains the main result in the paper, but it is inscrutable - maybe try a box plot or violin plot to aggregate results.

*Discussion*

Line 166: This sentence needs to be rephrased or broken up in two: *If both very specific and very broadly used regions...*

---

### Official Review · Reviewer_RpZf · 2021-10-27
**Interesting work but more control experiments would serve the quality of the paper**

**Rating:** 6
**Confidence:** 3

**Review:**

### Summary :
In this paper the authors explore the branch specialization effect in neural network in order to better understand branch specialization in neuroscience. To do so, the authors design 2 types of experiment, each with different network architectures and different tasks. In the first experiment, the authors studied the impact of architectural biases (in the form of the network architecture) on angle and color prediction tasks. They conclude that the network allocate the task to the branch with an appropriate inductive bias. In the second experiment the authors demonstrate that even without particular inductive biases, the sub-networks tend to specialize (for angle and size prediction tasks). Finally, the authors demonstrate that the branch specialization could be controlled by curriculum learning.

### General comments :
The article is in general well written. I found interesting to minimize the complexity of the tasks (with Gabor filters as input image) to better decipher the mechanism underlying the branch specialization. Nevertheless some of the claim made by the authors (mostly the claim on the impact of the inductive bias) might deserve more control experiment to be validated. For example, It would be interesting to better assess the impact of the number of parameters in each branch on specialization. The second part of the article is clearer and the experiments seem to be relevant. Please find below more detailed comments. My rating : 6.

### More specific comments :
* In section 2, I was surprised by the sentence : ‘We expected the angle task to be better suited to the inductive biases of the convolutional branch » (line 63). I fully understand the intuition behind this sentence (i.e. local kernel are better suited to grasp angle differences) but in the same time I consider convolutional network as a particular case (with constraints as weight sharing, and local kernel) of fully connected networks. My hypothesis is that this is more the difference in the number of parameters than the architecture biases in both branches that makes the difference (265 k parameters for the ConvNet versus 89k parameters in the FC network). To test such an hypothesis it would be interesting to re-run experiment 1 while varying the number of parameters in the FC network.

* For all experiments, it would be interesting to compare  a single branch network with the two branches version of your network. Such an experiment would allow to verify more rigorously that some architecture biases are more relevant for some tasks than other (e.g. ConvNet are better suited for angle prediction than for color prediction tasks).


### Typos:
Line 79, isn’t figure 2b (lower panel) rather than figure 2c ?

---

### Official Review · Reviewer_hwrw · 2021-10-27
**Can ANNs develop branch specialization?**

**Rating:** 5
**Confidence:** 4

**Review:**

### Summary:
This paper examined the ability of artificial neural networks (ANNs) in branch specialization. With an ANN with two parallel branches trained on two different tasks based on one single input image, does each branch learn different specializations? To answer this question, the authors trained a neural network with two branches on angle estimation and average color or size estimation based on images of Gabor patches as inputs. They showed that with and without inductive biases, the two branches did become specialized, and the specialization could be controlled via curriculum learning.

I believe the results presented in this paper are very important and, along with other recent findings, can raise interesting discussions and follow up studies in both vision neuroscience and machine vision communities. A few recent studies [1]-[2] also showed the importance of branched architectures in modeling the visual system. However, there is one major issue that, I believe, needs to be addressed to make the claims more reliable. I will explain it below.

### Major comments:

- In the experiments with the dual-task training, it could be imagined that the models reached a good accuracy just by optimizing the final linear layer based on the output of the randomly initialized backbones. The authors need to show that the connection weights within the two branches also changed during learning. Otherwise, it could be that the optimization of the linear output layer underlied the improved performances, which then wouldn’t support the branch specialization hypothesis.

Some other questions:

- Why did the authors decide to work with size estimation as the second task for the experiment with the branched conv architecture? How would the results change if the avg color estimation, that was used in the first experiment, was also used as the second task in the second experiment?

- The reasoning behind the choice of the dual tasks is not very clear. The avg color estimation in figure 2 and size estimation in figure 3 are too simple. One can even argue that the avg color and size can be estimated simply from pixel representations. This is not the case for the angle estimation task though. Why didn’t the authors use a second task that was more related to the first task and also couldn’t be easily solved at pixel-level representation of images; examples would be spatial frequency estimation, or phase estimation of the Gabor images.

### Minor comment:

- The two grey colors used in figures 2,3,4 (both branches, both lesioned) cannot be distinguished especially with the overlapped dashed (blue/green) lines. Maybe consider using another color scheme.


### References:

[1] Nayebi et al. "Unsupervised Models of Mouse Visual Cortex." bioRxiv (2021).

[2] Bakhtiari et al. "The functional specialization of visual cortex emerges from training parallel pathways with self-supervised predictive learning." bioRxiv (2021).

---

### Official Review · Reviewer_Q3bM · 2021-10-29
**Towards understanding branch specialization: some interesting first steps**

**Rating:** 7
**Confidence:** 4

**Review:**

The paper "What Matters in Branch Specialization? Using a Toy Task to Make Predictions" attempts to provide some empirically-grounded understanding of specialization in branching networks, where processing is split into two or more streams that do not communicate but then project onto common task representations. Previous literature has shown that such networks tend to specialize, but have typically considered networks where branches have identical architectural inductive biases and only differ in their random initialization or possibly downstream connectivity. Moreover, work cited by the authors using branching networks has used more complex tasks (e.g. naturalistic object and face recognition) which are more difficult to study. Here, the authors use tasks involving gabor wavelets, tasks which are simple and easy to align with the inductive biases of specific architectures, and quick to run many experiments on.

Pros:
1) The first main finding is that the inductive bias of a branch can predictably determine the learned specialization -- a convolutional branch is better at learning a feature detection task, and a fully connected branch is better at learning to regress a simple global signal. These results are not surprising but it is nevertheless interesting to seem them quantified. The discussion of how localized architectural inductive biases in the brain may lead to consistent anatomical specialization in the ventral stream is interesting and relevant.
2) The second main finding is that for identical branches and tasks with more similar inductive biases, branch specialization happens consistently. This can be expected from previous work but again the quantification is useful.
3) The third main finding is that by alternating tasks over longer durations -- rather than intermixing training -- branch specialization can be reduced. I think this is the most novel finding of the paper.

Cons/points for improvement:
1) The authors appear to consider branching networks as a novel architecture, but it is unclear how these networks are fundamentally different from the classic mixture of experts networks (Jacobs, Jordan, Nowlan, Hinton, 1991), which are not cited in this work. The authors may do well to survey the mixture of experts literature for potentially similar work in examining architectural inductive biases and other factors influencing learned specialization.
2) In section 2, can the authors justify why they chose to set up the FC branch to have 3x fewer parameters than the convolutional branch? Of three possible choices (this, matched parameters, matched number of channels and thus greater number of parameters for FC branch), this seems the least justified. If there is no real justification, I would suggest that the authors redo their analyses with matched # of parameters in the two branches.
3) It would aid visualization to match the y-axis limit for Figure 2b,c and 3b,c.
4) It was surprising to see how little a given branch contributed to the unmatched task when loss was scaled entirely towards that task. It would be useful to know how well each branch performs at baseline for both tasks (in particular, the one it tends not to specialize for) when trained from scratch with the other branch lesioned (in this case, it should actually learn the task to some extent). Alternatively, the authors could do an equivalent convex combination parameter exploration for scaling gradient updates in each branch to see how that affects the competition.
5) Figure 4c is a bit difficult to view due to the overlap of data points - i would recommend a greater shift between dots of a given group or switching to a grouped bar plot.
6) Related to pro 3), it should be noted that the reduction in specialization in a slow task-switching network corresponds to a decrease in overall performance. Thus, I am not sure that this mechanism is a reasonable hypothesis for the domain generality of the multiple demand network, which is thought to underly humans flexible (and successful) ability to process a wide variety of information.
7) Related to pro 1), the authors should probably cite Weiner & Grill-Spector 2014 and more recent works by the Weiner group for the claim that architectural biases in the ventral stream may determine anatomical localization of specialization. Also, the authors might consider noting that an equally plausible source of specialization is not the inductive bias of the internal architecture of the branch, but the inductive bias of the upstream and downstream projections of a branch. In other words, connectivity may determine specialization (e.g. Mahon and Caramazza, 2011; Plaut & Behrmann, 2011).

---

### Decision · Program_Chairs · 2021-11-02

Accept (Poster)